# BACKGROUND MATTERS TOO:
# A LANGUAGE-ENHANCED ADVERSARIAL FRAMEWORK FOR PERSON RE-IDENTIFICATION

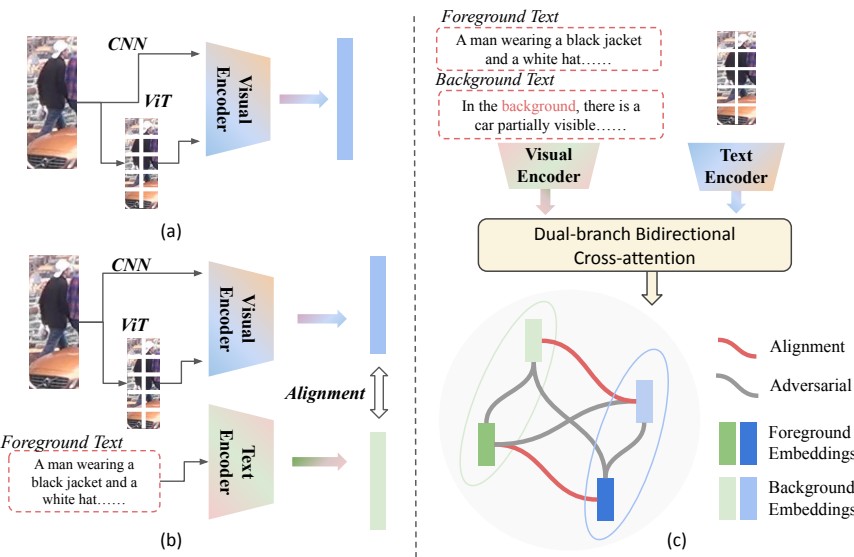

Figure 1: Frameworks of different person re-identification methods. (a) A framework purely based on a vision model. (b) A framework combining a vision model and a language model to align global or local image and text embeddings. (c) Our proposed framework, which is built upon intra-semantic alignment and inter-semantic adversarial learning.

## ABSTRACT

Person re-identification faces two core challenges: precisely locating the foreground target while suppressing background noise and extracting fine-grained features from the target region. Numerous visual-only approaches address these issues by partitioning an image and applying attention modules, yet they rely on costly manual annotations and struggle with complex occlusions. Recent multimodal methods, motivated by CLIP, introduce semantic cues to guide visual understanding. However, they focus solely on foreground information, but overlook the potential value of background cues. Inspired by human perception, we argue that background semantics are as important as the foreground semantics in ReID, as humans tend to eliminate background distractions while focusing on target appearance. Therefore, this paper proposes an end-to-end framework that jointly models foreground and background information within a dual-branch bidirectional cross-attention feature extraction pipeline. To help the network distinguish between the two domains, we propose an intra-semantic alignment and inter-semantic adversarial learning strategy. Specifically, we align visual and textual features that share the same semantics across domains, while simultaneously penalizing similarity between foreground and background features to enhance the network's discriminative power. This strategy drives the model to actively suppress noisy background regions and enhance attention toward identity-relevant

foreground cues. Comprehensive experiments on two holistic and two occluded ReID benchmarks demonstrate the effectiveness and generality of the proposed method, with results that match or surpass those of current state-of-the-art approaches.

# 1 INTRODUCTION

Person Re-identification (ReID) aims to match the same person across different scenes and camera views. Due to environmental complexity, the main challenges arise from diverse viewpoints, pose variations, background interference, and occlusions Ning et al. (2024). Addressing these challenges boils down to a single objective: learning fine-grained feature representations that are both robust to noise and invariant to diverse perturbations. More precisely, (1) how to locate the target's foreground while ignoring background disturbances; and (2) how to extract fine-grained features from that foreground region. Extensive efforts have been made to achieve this objective, which can be grouped into two major directions: model-driven and data-driven approaches Tan et al. (2024).

Model-driven methods focus on local feature learning, employing strategies such as pose estimation Wang et al. (2020), body segmentation Huang et al. (2025); Kim et al. (2022), semantic segmentation Gao et al. (2020); Zhu et al. (2020), and attention modules Mao et al. (2023) to provide structured body-part cues. While effective in regular scenarios, these tools are typically limited to spatial segmentation and require manual annotations, making them less adaptive to irregular or unseen occlusions. Meanwhile, data-driven methods aim to construct occlusion-enhanced data from existing datasets or manually generated samples Tan et al. (2024); Xia et al. (2024); Wu et al. (2024). However, manually introducing occlusions cannot deal with complex or previously unseen scenarios, such as irregular objects like trees and handrails.

Some other methods explicitly differentiate foreground from background and apply a mask to indicate the target region Liu et al. (2021); Yang et al. (2023); Liu et al. (2022). However, these methods do not genuinely understand the semantic content of images as humans do, and they are likely to miss the visual cues critical for identity recognition. *What do we, as human beings, think when we encounter someone we've met before?* The initial process involves filtering out background distractions while concurrently retrieving from memory visual cues of similar appearance, which aligns well with the challenges described above.

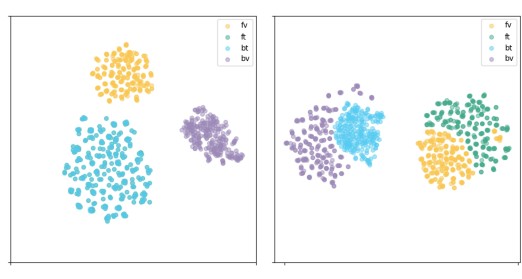

(a) w/o diversity loss      (b) w/ diversity loss

Figure 2: t-SNE visualization of cross-modal features, where each point represents an image feature. (a) Without diversity loss: 'f' and 'b' (foreground and background) features are entangled and 'v' and 't' (visual and text) are misaligned. (b) With diversity loss: 'v' and 't' features are well aligned, and 'f' and 'b' are clearly separated.

Multimodal supervision provide a possible solution to capture semantic cues beyond rigid masks, enabling better generalization to irregular and unseen scenarios. CLIP-ReID Li et al. (2023), as the pioneer in aligning semantic and visual information in ReID, has achieved great performance in some benchmarks. Inspired by this success, more and more image-text based methods have been proposed to achieve better token-level and feature-level alignment Yan et al. (2023); Jiang & Ye (2023); Wang et al. (2023); Yang & Zhang (2024); Wu et al. (2024). However, existing methods focus solely on modeling the foreground region and neglect the potentially informative background context. We argue that incorporating background information helps the model better distinguish semantic cues between foreground and background, reducing feature confusion when constraints or alignments are applied solely to the target region. Additionally, providing a semantic logit for the background enables the model to handle unseen or irregular background objects and textures that a purely visual pipeline would overlook.

In this paper, we propose FBA (**F**oreground and **B**ackground **A**dversarial Person Re-identification), a language-enhanced end-to-end framework that emulates human perception by jointly modeling

foreground and background information in a cross-modal feature-extraction pipeline. To capture precise semantic cues of different components in the image and guide attention toward the target regions, the proposed framework is built upon **intra-semantic alignment** and **inter-semantic adversarial learning**, as shown in Fig. 1. The former adopts an alignment strategy similar to CLIP Radford et al. (2021) to capture fine-grained multimodal features, while the latter introduces a diversity loss to help the model distinguish foreground targets from background distractions under the guidance of both visual and semantic information. Fig. 2 visualizes the feature distances across different domains, providing insight into our strategy.

Unlike CLIP-based approaches that simply align global visual and text embeddings after encoding, we perform local patch-to-prompt intersections similar to Yang et al. (2024) but employing a dual-branch bidirectional cross-attention mechanism with four weight-shared cross-attention modules to associate image patches and prompt tokens belonging to the same semantic group. A shared four-layer self-attention module and feed-forward network further enhance both global and local fine-grained feature extraction. Following Kang et al. (2025), who demonstrate that CLIP's latent space lacks compositional expressivity, we retain all patch and token embeddings for local intersections. An attention map differential pooling strategy is then applied to filter out non-informative embeddings. The overall structure of our network is shown in Fig. 3.

The main contributions of this work could be summarized as follows:

- We propose FBA, an end-to-end dual-branch bidirectional cross-attention framework that treats both foreground and background semantics as equally important. It employs local patch-to-prompt interactions to capture fine-grained representations and to build human-level understanding of the image.

- We introduce an intra-semantic alignment and inter-semantic adversarial learning strategy that aligns semantically consistent multimodal features and penalizes the feature distance between distractor and target regions.

- Our model achieves competitive results on both holistic and occluded ReID datasets, demonstrating its strong performance and generalization capability.

## 2 RELATED WORKS

### 2.1 PERSON RE-IDENTIFICATION

Person re-identification has been studied for a long time, yet several critical challenges remain to be addressed. The primary challenges arise from diverse viewpoints, pose variations, noise interference, and occlusions. More generally, how to extract fine-grained features from the target region across multiple cameras while remaining robust to noise is still challenging. To capture local features, PCB Sun et al. (2018) partitions the feature map into fixed horizontal stripes and learns independent part-level descriptors for each stripe. HOReID Wang et al. (2020) utilizes high-order mapping of multilevel feature similarities to achieve fine-grained semantic pose alignment. ISP Zhu et al. (2020) generates pseudo pixel-level labels for both body parts and personal belongings, then extracts local features of only the visible regions. PAT Li et al. (2021) employs an transformer encoder-decoder to discover diverse part prototypes via pixel-context encoding and prototype-based decoding for robust occluded person ReID. TransReID He et al. (2021), the first pure ViT-based ReID approach, extracts both global and local features with a jigsaw patch module and employs side-information embeddings to mitigate camera/view bias. With the powerful global receptive field, an increasing number of ViT-based methods have been proposed Zhu et al. (2022); Tan et al. (2022); Zhu et al. (2023); Xia et al. (2024). While these methods depend on improved feature alignment, other studies have adopted data-driven strategies, such as random erasing Zhong et al. (2020) and manually generated samples Wang et al. (2022a;b); Tan et al. (2024); Xia et al. (2024); Wu et al. (2024).

Other approaches explicitly distinguish between foreground and background, employing a mask to delineate the target region. FA-Net Liu et al. (2021) introduces an end-to-end branch that explicitly localizes pedestrian foregrounds and extracts foreground-focused features. F-BDMTrack Yang et al. (2023) employs fore-background distribution-aware attention within a transformer architecture to robustly discriminate targets and suppress background. However, these approaches lack human-level

semantic understanding of images and may overlook visual cues in background that are essential for identity recognition.

## 2.2 Vision-Language Pre-training

Recent work shows that large-scale vision–language pre-training (VLP) provides Re-ID models with richer semantic priors than image-only pre-training. General-purpose VLP models such as CLIP Radford et al. (2021) learn aligned image–text representations, enabling the injection of textual cues (for example, clothing color or carried objects) that are difficult to capture using only RGB images. The pioneering CLIP-ReID Li et al. (2023) adopts a two-stage strategy. In the first stage, it trains a set of learnable text tokens for each image, similar to prompt learning in CoOp Zhou et al. (2022). In the second stage, it optimizes the visual encoder. Inspired by it, various works have been focused on VLP based text-to-image person retrieval tasks. CFine Yan et al. (2023) utilizes CLIP's rich multi-modal knowledge to mine fine-grained intra-modal and inter-modal discriminative clues. IRRA Jiang & Ye (2023) adds random masking to text tokens and employs a token classifier after the cross attention layers to mine fine-grained global representations. TP-PS Wang et al. (2023) further explores the modality association by constraints of various integrity and prompts for attribute hints. MGCC Wu et al. (2024) applies a token selection mechanism to filter out non informative tokens and then feeds the remaining tokens into a global and local contrastive consistency alignment module. However, these methods restrict cross modal alignment to foreground features at both global and local levels, overlooking background semantic cues and the interaction between foreground and background that humans naturally use in similar recognition tasks.

## 3 Preliminary

### 3.1 Overview of CLIP-based Methods

Before introducing our proposed framework, we briefly review the CLIP-based methods. The CLIP and CLIP based methods align the visual and textual embeddings (namely class [CLS] and end [EOS] tokens) after visual encoder $\mathcal{V}(\cdot)$ and text encoder $\mathcal{T}(\cdot)$ with contrastive loss as in Radford et al. (2021). The image-to-text contrastive loss is calculated as:

$$\mathcal{L}_{i2t}(i) = -\log \frac{\exp(s(\mathbf{v}_i, \mathbf{t}_i))}{\sum_{j=1}^{B} \exp(s(\mathbf{v}_i, \mathbf{t}_j))} \tag{1}$$

where $(\mathbf{v}_i, \mathbf{t}_i)$ denote the visual and textual embeddings of the $i$-th matched pair, $s(\cdot, \cdot)$ is the similarity function (e.g., cosine similarity), and $B$ is the batch size. The text-to-image contrastive loss shares the similar form. Derivative methods based on CLIP further address its limitation in handling multiple images of the same identity, which should ideally share similar textual descriptions Li et al. (2023); Yang et al. (2024).

Currently, many approaches focus on improving modality alignment and exploring the representational potential of CLIP in both vision and language. However, these methods often overlook the intuitive strategy adopted by humans in similar tasks, which involves first distinguishing between foreground and background. Although some methods based on masks or attention mechanisms have considered this characteristic, they do not effectively exploit the rich semantic information contained in the image, especially the background semantics that are present but usually ignored.

## 4 Methodology

To address the aforementioned gap, we propose FBA, an adversarial ReID framework that fully leverages interactions between foreground and background visual–semantic information to mimic human perception. It also exploits the great potential of CLIP embeddings.

### 4.1 Multimodal Representation Encoding

To effectively capture both visual and textual information, we design a dual-stream encoding framework. The visual encoder $\mathcal{V}(\cdot)$ is trainable, whereas the text encoder $\mathcal{T}(\cdot)$ remains frozen to supply

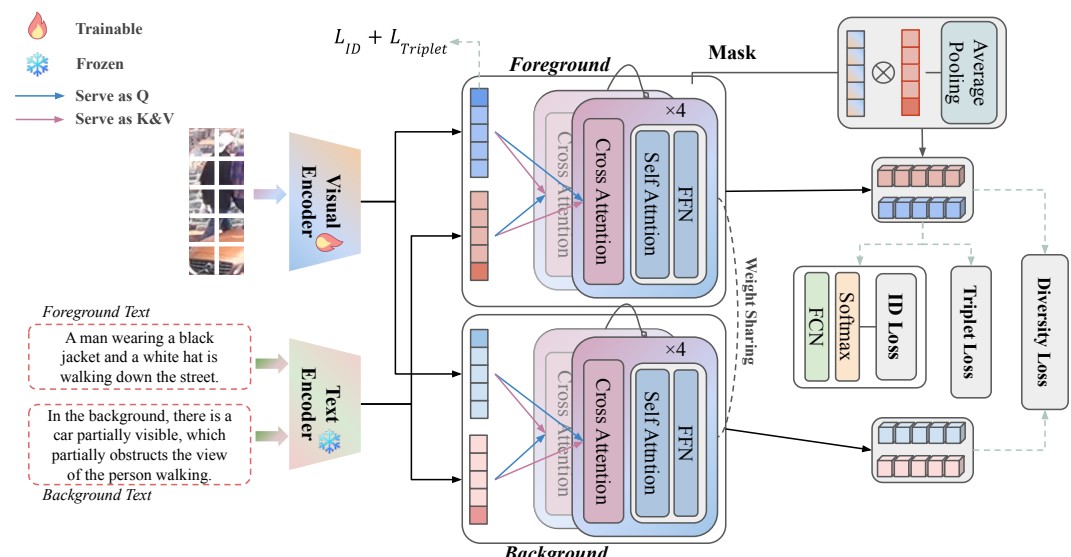

Figure 3: Overview of the proposed Foreground and Background Adversarial Image-Text Person Re-Identification framework (FBA). Embeddings from the visual and text encoders are fed into a dual-branch bidirectional cross-attention module, with one branch dedicated to the foreground and the other to the background. Each branch operates in two directions, where either visual or textual embeddings can serve as *Query*, enabling the model to capture fine-grained cues from different components. The optimization is performed through intra-semantic alignment and inter-semantic adversarial learning, which forces the model to actively suppress noisy background regions and enhance attention toward identity-relevant foreground cues. Furthermore, an attention map differential pooling strategy is proposed to enhance informative patch selection.

stable language priors. The foreground and background textual descriptions are generated from a large language model Liu et al. (2023) with the prompts:

> **Foreground Prompt:** "*Describe the appearance of persons in the image, focusing on their appearance, attire and accessories.*"

> **Background Prompt:** "*Describe the background in the image with less than 50 words, focusing on any objects or elements that might obscure the view of the person.*"

The foreground text describes the target identity and background text provides contextual information. These embeddings are then forwarded into the cross-modal interaction module.

## 4.2 DUAL-BRANCH BIDIRECTIONAL CROSS-ATTENTION MODULE

To disentangle foreground and background semantics, we employ two parallel transformer branches, each consisting of a cross-attention module and four stacked self-attention and FFN modules. The outputs from the encoders are divided into two groups, $\{V_f, T_f\}$ and $\{V_b, T_b\}$, which are then fed into the dual-branch cross-attention module. The subscripts $f$ and $b$ denote the foreground and background, respectively. $V_{i\in(f,b)} = [[CLS], \mathbf{v}_i^1, \mathbf{v}_i^2, ..., \mathbf{v}_i^N]$, where $N$ is the number of image patches. $T_{i\in(f,b)} = [[SOS], \mathbf{t}_i^1, \mathbf{t}_i^2, ..., \mathbf{t}_i^M, [EOS]]$, where $M$ is the number of text tokens. In the cross-attention module, most existing methods adopt image embeddings solely as *Key* and *Value* in cross-modal attention. In contrast, our method also utilizes image embeddings as *Query*, enabling a bidirectional interaction where both modalities can attend to each other more effectively. At the same time, all patches and tokens are preserved to fully exploit the fine-grained local cues. In summary, we adopt four weight shared Transformer blocks with two for foreground intersection and two for background.

The $[CLS]$ and $[EOS]$ tokens are then selected from the outputs of the cross-attention module, specifically $\{\mathcal{F}_f^T, \mathcal{F}_f^V\}$ for the foreground branch and $\{\mathcal{F}_b^T, \mathcal{F}_b^V\}$ for the background branch. The superscripts $V$ and $T$ indicate that vision or text is used as the $Query$, respectively. Before computing the loss, the foreground text-guided embeddings are processed using a pooling strategy introduced in the following section.

## 4.3 ATTENTION MAP DIFFERENTIAL POOLING

An attention map differential pooling strategy is proposed to further explore useful cues across all token embeddings, rather than focusing solely on the most critical ones. This idea is inspired by the work in Kang et al. (2025). With the aim of assigning more attention to tokens that better distinguish foreground from background, we first calculate the attention weights $\mathcal{W}_f \in \mathbb{R}^{N \times M}$ from the cross-attention layer of the foreground branch and $\mathcal{W}_b \in \mathbb{R}^{N \times M}$ from the background branch. To quantify the discrepancy between these two attention maps, we compute the cosine similarity between their corresponding column vectors, resulting in an $M$-dimensional vector $\mathbf{s} \in \mathbb{R}^{1 \times M}$:

$$s_j = \frac{\langle \mathcal{W}_f^{(:,j)}, \mathcal{W}_b^{(:,j)} \rangle}{\|\mathcal{W}_f^{(:,j)}\|_2 \cdot \|\mathcal{W}_b^{(:,j)}\|_2}, \quad j = 1, 2, \ldots, M \tag{2}$$

We then apply a min-max normalization to this similarity vector to generate a token-wise attention mask $\mathbf{w} \in \mathbb{R}^{1 \times M}$:

$$w_j = 1 - \tilde{s}_j = 1 - \frac{s_j - \min(\mathbf{s})}{\max(\mathbf{s}) - \min(\mathbf{s}) + \varepsilon} \tag{3}$$

where $\varepsilon$ is a small constant for numerical stability and $w_j \in \mathbf{w}$. This normalized mask is then used to aggregate the foreground text guided token embeddings as a pooling strategy:

$$\bar{\mathcal{F}}_f^T = \sum_j \alpha_j f_j, \quad \alpha_j = \frac{\exp(\tau w_j)}{\sum_k \exp(\tau w_k)} \tag{4}$$

where $f_j$ is the j-th foreground text-guided embedding and $\tau$ is the temperature. To avoid ambiguity, we simply write $\bar{\mathcal{F}}_f^T$ as $\mathcal{F}_f^T$ hereafter. In other words, by focusing on tokens that exhibit stronger identity-related attention across foreground and background, the final feature representation incorporates richer identity cues.

## 4.4 OBJECTIVE FUNCTION

The overall framework is optimized using a combination of identity classification (ID) loss and triplet loss to ensure discriminative feature learning. These two losses are applied only to the foreground-guided features $\mathcal{F}_f^T$ and $\mathcal{F}_f^V$, and the features generated from the visual encoder backbone.

$$\mathcal{L}_{\text{ID}} = -\frac{1}{B} \sum_{i=1}^{B} y_i \log \hat{y}_i \tag{5}$$

$$\mathcal{L}_{\text{Triplet}} = \max \left( \|f_a - f_p\|_2^2 - \|f_a - f_n\|_2^2 + m, \, 0 \right) \tag{6}$$

In the ID loss, $\hat{y}_i$ denotes the predicted probability of the ground-truth class $y_i$, and $B$ is the batch size. In the triplet loss, $f_a$, $f_p$, and $f_n$ are the feature embeddings of the anchor, positive, and negative samples, respectively. The margin $m$ controls the minimum distance between positive and negative pairs.

Additionally, to encourage feature diversity between foreground and background representations, we introduce a diversity loss $\mathcal{L}_{\text{div}}$ that penalizes inter-semantic (fore–back) similarity while promoting intra-semantic (fore–fore and back–back) alignment.

$$\mathcal{L}_{\text{tri-div}} = \sum_{a,b,c,d \in \mathcal{P}} \mathcal{L}_{\text{Triplet}}^{a,b|c,d} \tag{7}$$

$$\mathcal{L}_{\text{con}} = (1 - s(\mathcal{F}_f^T, \mathcal{F}_f^V)) + (1 - s(\mathcal{F}_b^T, \mathcal{F}_b^V)) \tag{8}$$

$$\mathcal{L}_{\text{div}} = \mathcal{L}_{\text{tri-div}} + \mathcal{L}_{\text{con}} \tag{9}$$

where $\mathcal{L}_{\text{Triplet}}^{a,b|c,d}$ denotes the triplet loss sum of two triplets $(a,b|a,c)$ and $(a,b|a,d)$. The set $\mathcal{P}$ contains all four complementary configurations between foreground and background features:

$$\mathcal{P} = \left\{ \begin{array}{l} (\mathcal{F}_f^T, \mathcal{F}_f^V \mid \mathcal{F}_b^T, \mathcal{F}_b^V), \\ (\mathcal{F}_b^T, \mathcal{F}_b^V \mid \mathcal{F}_f^T, \mathcal{F}_f^V), \\ (\mathcal{F}_f^V, \mathcal{F}_f^T \mid \mathcal{F}_b^T, \mathcal{F}_b^V), \\ (\mathcal{F}_b^V, \mathcal{F}_b^T \mid \mathcal{F}_f^T, \mathcal{F}_f^V) \end{array} \right\} \tag{10}$$

While $\mathcal{L}_{\text{tri-div}}$ enforces a multi-view triplet-based alignment of intra-semantic features and penalizes inter-semantic features, it primarily focuses on the relative distance between positive and negative samples. To further enhance the absolute consistency of modality-specific representations, we introduce an auxiliary contrastive loss $\mathcal{L}_{\text{con}}$, which directly encourages similarity between paired text and visual features within the same semantic region.

The overall objective function combines identity loss, triplet loss, and proposed diversity loss. The training objective is formulated as follows:

$$\mathcal{L}_{\text{total}} = \mathcal{L}_{\text{ID}} + \mathcal{L}_{\text{Triplet}} + \mathcal{L}_{\text{ID}}^C + \mathcal{L}_{\text{Triplet}}^C + \lambda \mathcal{L}_{\text{div}} \tag{11}$$

where the superscript $C$ denotes the loss calculated from cross-modal features. The hyperparameter $\lambda$ balances the contribution of the diversity loss. During inference, the model employs only the foreground-guided features, concatenated with those from the visual backbone.

## 5 EXPERIMENTS

### 5.1 DATASETS AND EVALUATION PROTOCOLS

The proposed method in this work is fully evaluated through four person re-identification datasets, including two holistic datasets DukeMTMC-reID Zheng et al. (2017) and CUHK03-NP (labeled) Li et al. (2014), and two occluded datasets Occluded-Duke Miao et al. (2019) and Occluded-ReID Zhuo et al. (2018). The details of those datasets are summarized in Appendix A.2. The Market-1501 dataset is excluded from our evaluation because its noisy annotations and detection errors can easily interfere with the stability of such adversarial training. Following established practices, we adopt mean Average Precision (mAP) and Rank-1 (R-1) accuracy as the primary evaluation metrics.

### 5.2 IMPLEMENTATION DETAILS

We adopt the ViT-B/16 model pre-trained by CLIP as our backbone with a sliding-window setting as in He et al. (2021). The visual backbone consists of 12 transformer layers with a hidden size of 768. A linear projection layer is used to map the 512-dimensional output of the text encoder to 768 dimensions. Foreground and background captions are generated using the LLaVA v1.5-7b model Liu et al. (2023). All input images are resized to $384 \times 128$, with a batch size of 64, comprising 16 identities and 4 images per identity. We use the Adam optimizer with a weight decay of 1e-4. The model is trained for 60 epochs, including 10 warm-up epochs during which the learning rate increases linearly from $0.001 \times$ base learning rate to the base learning rate, followed by a cosine decay to $0.01 \times$ base learning rate. To accommodate the varying scale and complexity of different datasets, we empirically set the base learning rate for each dataset (8e-5 for DukeMTMC-reID and Occluded-Duke, and 1.2e-4 for CUHK03-NP). For fairness, baseline methods were trained with their official hyperparameters or re-implemented under the same search range, and we found their results consistent with reported numbers. Note that the Occluded-ReID is used only as a test set. This per-dataset adjustment improves training stability and convergence performance. The margin $m$ for the triplet loss is set to 0.3, and the balance factor in Eq. (11) is $\lambda = 0.5$. The entire framework is implemented using PyTorch and trained on 4 NVIDIA A6000 GPUs.

### 5.3 COMPARISON WITH STATE-OF-THE-ART METHODS

This section compares our proposed method with several state-of-the-art approaches, including those based on CNN and ViT backbones. A detailed analysis of the results presented in Table 1 is provided below. More samples are visualized in Appendix A.1.

Table 1: Comparison of CNN- and ViT-based methods on DukeMTMC, CUHK03-NP (labeled), Occluded-Duke, and Occluded-ReID datasets.

| Backbone | Method | Reference | DukeMTMC | | CUHK03-NP | | Occluded-Duke | | Occluded-ReID | |
|---|---|---|---|---|---|---|---|---|---|---|
| | | | mAP | R-1 | mAP | R-1 | mAP | R-1 | mAP | R-1 |
| CNN | PCB | ECCV'2018 | 69.2 | 83.3 | 57.5 | 63.7 | - | - | - | - |
| | DSR | CVPR'2018 | - | - | - | - | 30.4 | 40.8 | 62.8 | 72.8 |
| | OSNet | ICCV'2019 | 73.5 | 88.6 | 67.8 | 72.3 | - | - | - | - |
| | HOReID | CVPR'2020 | 75.6 | 86.9 | - | - | 43.8 | 55.1 | 70.2 | 80.3 |
| | PVPM | CVPR'2020 | - | - | - | - | 37.7 | 47.0 | 59.5 | 66.8 |
| | ISP | ECCV'2020 | 80.0 | 89.6 | 74.1 | 76.5 | 52.3 | 62.8 | - | - |
| | PAT | CVPR'2021 | 78.2 | 88.8 | - | - | 53.6 | 64.5 | 72.1 | 81.6 |
| | ALDER | TIP'2021 | 78.9 | 89.9 | 78.7 | 81.0 | - | - | - | - |
| | Part-Label | ICCV'2021 | - | - | - | - | 46.3 | 62.2 | 71.0 | 81.0 |
| | LTReID | TMM'2022 | 80.4 | 90.5 | 80.3 | 82.1 | - | - | - | - |
| | DRL-Net | TMM'2022 | 76.6 | 88.1 | - | | 50.8 | 65.0 | - | - |
| | CLIP-ReID | AAAI'2023 | 80.7 | 90.0 | - | - | 53.5 | 61.0 | - | - |
| | PromptSG | CVPR'2024 | 80.4 | 90.2 | 79.8 | 80.5 | - | - | - | - |
| ViT | TransReID | ICCV'2021 | 82.0 | 90.7 | - | - | 59.2 | 66.4 | - | - |
| | FED | CVPR'2022 | 78.0 | 89.4 | - | - | 56.4 | 68.1 | 79.3 | 86.3 |
| | AAFormer | TNNLS'2023 | 80.9 | 90.1 | 79.0 | 80.3 | 58.2 | 67.1 | - | - |
| | CLIP-ReID | AAAI'2023 | 82.5 | 90.0 | - | - | 59.5 | 67.1 | - | - |
| | PromptSG | CVPR'2024 | 81.6 | 91.0 | 83.1 | 85.1 | - | - | - | - |
| | Baseline | | 80.0 | 88.8 | 82.9 | 84.8 | 53.5 | 60.8 | 74.0 | 76.3 |
| | FBA | Ours | 81.7 | 91.5 | 85.3 | 86.6 | 60.5 | 69.5 | 84.0 | 85.4 |

### 5.3.1 HOLISTIC REID

We compare the performance of FBA with several existing methods on two holistic person ReID datasets. On DukeMTMC, FBA achieves a notable improvement over the baseline (+1.7% mAP, +2.7% R-1), and surpasses previous state-of-the-art methods in terms of R-1 accuracy. This result suggests that the integration of foreground-background semantic information with visual features enhances the model's ability to handle more challenging samples. However, the overall effect of the adversarial learning appears limited in mAP. We speculate that this may be due to multiple individuals often appearing in the same image, leading to less precise descriptions.

On the CUHK03-NP benchmark, our approach surpasses all existing methods, achieving 2.2% mAP and 1.5% R-1 improvement over PromptSG Yang et al. (2024). This demonstrates that the introduction of multimodal interaction and the diversity loss effectively enhances the model's ability to distinguish between foreground and background content.

### 5.3.2 OCCLUDED REID

Although our method does not incorporate explicit mechanisms designed for occlusion handling, as seen in works such as HOReID Wang et al. (2020), ISP Zhu et al. (2020), PAT Li et al. (2021), FED Wang et al. (2022b), and AAFormer Zhu et al. (2023), we still evaluate it on two occluded ReID datasets. This is because the adversarial learning design in our method is intrinsically capable of suppressing interference from obstructions. On Occluded-Duke, FBA outperforms all other methods (+1.0% mAP, +1.4% R-1). For Occluded-ReID, which is a test-only dataset, we use the model trained on CUHK03-NP and obtain 84% mAP, surpassing previous records, with R-1 accuracy second only to FED. These results demonstrate the robustness and strong generalization capability of FBA, particularly in handling challenging occluded scenarios.

### 5.4 ABLATION STUDIES

To quantitatively analyze the contribution of each component in our approach, we conduct ablation studies on the DukeMTMC and CUHK03-NP datasets, as shown in Table 2. In the first row, the baseline model uses only the features extracted by the backbone visual encoder. In the second row, the cross-modal interaction is introduced, while only the foreground branch contributes to the loss

computation. Comparing the first and second rows, the introduction of cross-modal interaction leads to a noticeable improvement in R-1 accuracy. The third row introduces the diversity loss, which improves both mAP and R-1 accuracy, demonstrating its effectiveness in helping the model better distinguish the target region from noise. In the last row, we incorporate the attention map differential pooling strategy, which allows the model to aggregate information beyond a single token. At this stage, all components together constitute the complete FBA architecture.

Table 2: Ablation study on the effectiveness of each loss component on DukeMTMC and CUHK03-NP.

| Components | | | | DukeMTMC | | | | CUHK03-NP | | | |
|---|---|---|---|---|---|---|---|---|---|---|---|
| $\mathcal{L}_{\text{ID}} + \mathcal{L}_{\text{Triplet}}$ | $\mathcal{L}_{\text{ID}}^{C} + \mathcal{L}_{\text{Triplet}}^{C}$ | $\mathcal{L}_{\text{div}}$ | $Diff$ | mAP | R-1 | R-5 | R-10 | mAP | R-1 | R-5 | R-10 |
| ✓ | | | | 80.0 | 88.8 | 94.5 | 95.9 | 82.9 | 84.8 | 93.2 | 96.4 |
| ✓ | ✓ | | | 80.0 | 90.3 | 94.6 | 95.6 | 83.1 | 85.0 | 93.1 | 96.8 |
| ✓ | ✓ | ✓ | | 80.7 | 90.4 | 94.6 | 96.1 | 84.9 | 86.3 | 94.1 | 97.1 |
| ✓ | ✓ | ✓ | ✓ | **81.7** | **91.5** | **95.3** | **96.3** | **85.3** | **86.6** | **94.3** | **97.1** |

Additional experiments are provided in the Appendix: hyperparameter analysis in Appendix A.3, inference feature combinations in Appendix A.4, and network design studies in Appendix A.5.

## 5.5 QUALITATIVE ANALYSIS

To better demonstrate the effectiveness of our method in distinguishing foreground from background, we visualize the attention maps in Fig. 4 using samples from the Occluded-Duke and Occluded-ReID datasets. These samples contain various types of occlusions, such as vehicles, railings, and non-target pedestrians, which pose considerable challenges. The attention weights are directly extracted from the cross-attention layers.

Our method demonstrates a strong ability to understand the semantic structure of foreground and background regions and to distinguish target person from distracting elements. For instance, in the first column of Occluded-Duke, FBA effectively identifies and suppresses interference from vehicles in the scene, directing attention to key attributes of the target individual such as the coat, backpack, and hair. In the second column, despite the presence of a distracting pedestrian with a backpack in front of the target, our method accurately avoids the interference. In the third column of Occluded-ReID, the target person is occluded by multiple layers of metal railings. Remarkably, the model focuses precisely on the person, avoiding the railings, and is even able to capture the body parts visible between the bars.

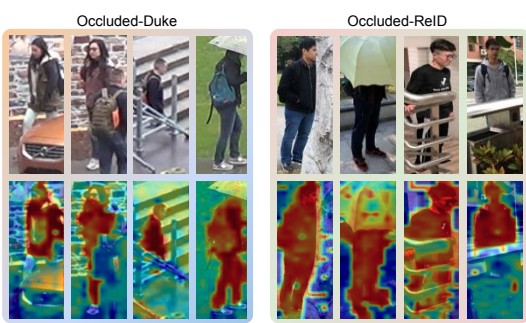

Figure 4: Attention map visualization of FBA on Occluded-Duke and Occluded-ReID. The first row represents the raw images and the second row presents the attention maps.

## 5.6 CONCLUSION

This paper proposes a language-enhanced end-to-end adversarial framework for person re-identification, named FBA. By employing a dual-branch bidirectional cross-attention module, FBA simultaneously models foreground and background semantics. A diversity loss and an attention map-based differential pooling strategy are further introduced to effectively distinguish target regions from background distractions. Experimental results show that FBA achieves or approaches state-of-the-art performance in terms of mAP and R-1 accuracy in both holistic and occluded person re-identification tasks. Future work will explore refined prompt engineering and lightweight model designs to improve large-scale, real-time deployment.

## REPRODUCIBILITY STATEMENT

The code of this paper is available at https://anonymous.4open.science/r/test-EFE84RCJ83U439.

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

# A APPENDIX

## A.1 MATCHING RESULTS ON DUKEMTMC (SAMPLES)

Fig. 5 shows sample matching results of the baseline and our method on DukeMTMC. In both occluded and non-occluded cases, our method achieves higher matching accuracy than the baseline.

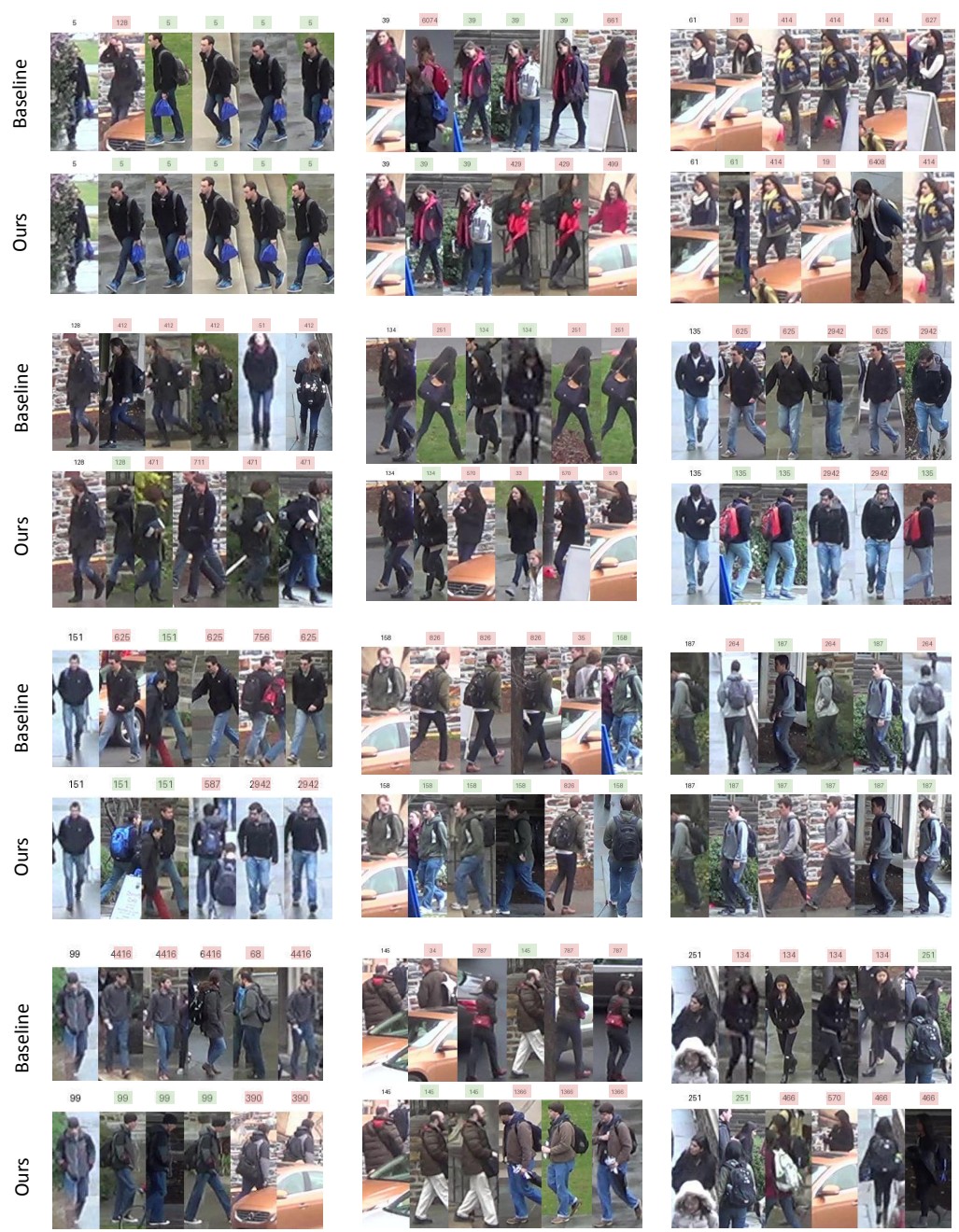

Figure 5: Sample matching results of the baseline and our method on DukeMTMC. The first image of each identity is the gallery image, followed by the top-5 matches. Red labels indicate incorrect matches, while Green labels indicate correct ones.

## A.2 STATISTICS OF THE DATASETS

Table 3: Statistics of ReID datasets in our experiments.

| Dataset | IDs | Images | Cams |
|---|---|---|---|
| DukeMTMC-reID | 1,404 | 36,411 | 8 |
| CUHK03-NP | 1,467 | 13,164 | 2 |
| Occluded-Duke | 1,404 | 35,489 | 8 |
| Occluded-ReID | 200 | 2,000 | - |

## A.3 ABLATION STUDY ON IMAGE SIZE AND STRIDE SIZE

We compare the performance of FBA on the CUHK03-NP dataset under different image sizes and stride settings, as shown in Table 4. When increasing the image resolution from $256 \times 128$ to $384 \times 128$, the R-1 accuracy remains nearly unchanged, while the mAP improves by 0.5%. A reduction in stride size from 16 to 12 notably improves performance, yielding a 1.3% increase in mAP and a 1.3-1.4% rise in R-1 accuracy. This suggests that a smaller stride enables the model to capture more fine-grained features. Notably, even under the least optimal configuration, our method still outperforms other approaches.

Table 4: Ablation analysis of different image and stride sizes during training on CUHK03-NP.

| Image Size | Stride Size | mAP | R-1 |
|---|---|---|---|
| 256×128 | 16 | 83.5 | 85.3 |
| 384×128 | 16 | 84.0 | 85.2 |
| 256×128 | 12 | 84.8 | 86.6 |
| 384×128 | 12 | 85.3 | 86.6 |

## A.4 ABLATION STUDY ON INFERENCE SETTINGS

Table 5 presents different feature combinations used during inference. $\mathcal{F}_{base}$ denotes the feature from visual backbone, which is trainable. $\mathcal{F}_{cross}^f$ and $\mathcal{F}_{cross}^b$ represent the cross-attention features from the foreground and background branches, respectively. Under the adversarial training mechanism, the foreground branch achieves much better performance than background since it focuses more on target regions, while the backbone also learns to capture fine-grained foreground cues after training. It is worth noting that on Occluded-Duke, where occlusions are more complex, combining $\mathcal{F}_{base}$ with $\mathcal{F}_{cross}^f$ yields better accuracy in challenging cases (as reflected in R-1). In contrast, on the relatively clean CUHK03 dataset, the trained backbone alone is already sufficiently powerful.

Table 5: Ablation study on different feature combinations during inference.

| Features | | | Occluded-Duke | | CUHK03-NP | |
|---|---|---|---|---|---|---|
| $\mathcal{F}_{base}$ | $\mathcal{F}_{cross}^f$ | $\mathcal{F}_{cross}^b$ | mAP | R-1 | mAP | R-1 |
| ✓ | | | 60.6 | 69.1 | 85.4 | 86.6 |
| | ✓ | | 54.8 | 64.3 | 80.6 | 82.9 |
| | | ✓ | 28.3 | 45.9 | 61.7 | 67.9 |
| ✓ | ✓ | | 60.5 | 69.5 | 85.3 | 86.6 |

## A.5 ABLATION STUDY ON NETWORK DESIGN

Table 6 presents the evaluation results of different cross-attention designs. Using text alone as the query ($G_T$) outperforms using visual features as the query ($G_V$), indicating that text-guided cross-attention is more effective in capturing target features. The bidirectional cross-attention ($G_V +$

$G_T$) further improves performance on both datasets, demonstrating the complementarity of the two queries and their ability to better model complex scenarios.

Table 6: Ablation analysis of bidirectional vs. unidirectional cross-attention.

| Datasets | Methods | mAP | R-1 |
|---|---|---|---|
| Occluded-Duke | $G_V$ | 54.2 | 62.7 |
| | $G_T$ | 59.0 | 67.9 |
| | $G_V + G_T$ | 60.5 | 69.5 |
| CUHK03-NP | $G_V$ | 82.4 | 83.4 |
| | $G_T$ | 84.1 | 85.6 |
| | $G_V + G_T$ | 85.3 | 86.6 |

## A.6 The Use of Large Language Models

In this work, large language models (LLMs) were only employed to assist in minor language polishing and grammatical refinement. All ideas, analyses, models, and experiments were fully deployed and implemented by the authors, while LLMs served solely as a tool for improving clarity and readability of the text.

