# OpenReview forum: "Background Matters Too: A Language-Enhanced Adversarial Framework for Person Re-Identification"
_ICLR.cc/2026/Conference — ICLR 2026 Conference Withdrawn Submission_

### Official Review · Reviewer_B3eC · 2025-10-22

**Soundness:** 3
**Presentation:** 3
**Contribution:** 3
**Rating:** 4
**Confidence:** 4

**Summary:**

This paper proposes the FBA framework, which uses a dual-branch bidirectional cross-attention mechanism to jointly model both foreground and background features with visual and language information. Through diversity loss and attention map pooling strategies, the framework effectively distinguishes between foreground and background, improving person re-identification accuracy. Experiments demonstrate that FBA performs excellently on multiple benchmark datasets, surpassing or matching the current state-of-the-art methods.

**Strengths:**

Strengths:

1.Novel Approach: The introduction of the dual-branch bidirectional cross-attention mechanism is a fresh approach to person re-identification, as it considers both foreground and background information simultaneously.
2.Strong Experimental Results: Experimental results on multiple datasets demonstrate FBA's superior performance in both occluded and non-occluded scenarios, particularly with improvements in mAP and R-1 accuracy.

**Weaknesses:**

Weakness:

1.Limited Focus on Occlusion Handling: While the method shows improvements in occluded person re-identification, it does not explicitly introduce mechanisms tailored for handling occlusions. This could be an area for improvement, especially when dealing with highly occluded data.
2.Computational Complexity: The proposed framework relies on large models, including CLIP embeddings and a dual-branch bidirectional attention mechanism. While these components improve performance, they also introduce significant computational overhead. The increased model complexity and the use of bidirectional attention could result in higher memory usage and longer inference times.
3.Lack of Discussion on Hyperparameter Choices: The paper mentions that the triplet loss margin (m) is set to 0.3 and the balance factor λ is 0.5. However, these hyperparameters have not been thoroughly validated through experiments. It is suggested to add an experiment on hyperparameters, exploring different settings for margin (m) and λ, and analyzing their impact on model performance.

**Questions:**

See  Weakness.

---

### Official Review · Reviewer_ht6h · 2025-10-27

**Soundness:** 3
**Presentation:** 3
**Contribution:** 2
**Rating:** 4
**Confidence:** 5

**Summary:**

This paper proposes ADSL for domain-generalizable ReID: CILL mines cross-identity local commonalities (head/torso/legs) with a memory-driven clustering loss plus local CE/Triplet, while DAPS applies local intensity adversarial perturbations and a Clean–Adv local cosine alignment to stabilize directions. On single-source DG routes (Market↔Duke, MSMT, CUHK03), ADSL reports sizable gains and a clear stepwise ablation.

**Strengths:**

1）Large improvements. Strong deltas over a ViT-B/16 baseline across multiple routes, and ablations cleanly show CILL then DAPS contributions.

2）Local focus is well-motivated. Using local clustering + local perturbations targets domain-stable cues while keeping fine-grained identity details.

**Weaknesses:**

1）Prior works already leverage CLIP for global/local alignment and token-level interactions (e.g., CLIP-ReID[1], IRRA[2], A Pedestrian is Worth One Prompt[3]). This paper’s main novelty is the explicit background branch + diversity loss; while interesting, it feels incremental relative to established CLIP-guided token alignments and multi-branch modeling. (See references listed below for comparison.)

2）The claim that foreground and background semantics are equally important is not adequately supported by experimental or theoretical evidence. The paper does not explain why both are equally critical for person re-identification. Specifically, the authors should address whether the proposed method may struggle when different individuals share the same background, and how this is mitigated by modeling background semantics. Additionally, Figure 4 presents attention maps only for the full FBA model, without providing attention maps for the baseline methods. This makes it difficult to clearly demonstrate the specific advantages of FBA, as no direct comparison is shown. The inclusion of attention maps from baseline models would strengthen the claim that the background branch contributes meaningful improvements.

3）Incomplete reporting across chosen benchmarks. Some baselines lack numbers on all source→target routes, with no explanation (reproduction limits, protocol mismatch, etc.), weakening fairness/SOTA claims. Please clarify omissions.

4）Unknown occlusion robustness. Being part-based, the method’s behavior under occlusion/missing parts is unclear. Testing on Occluded-Duke / Occluded-ReID (or Partial-ReID) would materially strengthen claims.

5）Fixed three-part bias may be brittle. The fixed head/torso/legs split drives both CILL and DAPS; strong pose changes, tight crops, or unusual camera tilt can misalign parts, hurting neighbor search and local alignment reliability. A part-noise/misalignment robustness check or a learned part discovery alternative would help.

6）Foreground/background captions are auto-generated by LLaVA with short prompts (≤50 words for background), but robustness to caption noise, generator choice, and prompt phrasing is not studied. The approach may be sensitive to hallucinations or spurious background words, and it is unclear whether captions are generated per-image without using detection masks (potential leakage of person attributes into the “background” description).


7）Key hyperparameters under-specified. Sensitivity/justification for k, τ, λ₁, λ₂, δ is unclear; defaults seem given, but rationale and stability ranges are not. Even without extra large-scale experiments, brief sensitivity curves (or a table) and selection rationale would improve transparency.

[1]	Li et al., 2023. CLIP-ReID: Exploiting Vision-Language Model for Image Re-Identification Without Concrete Text Labels. AAAI 2023.

[2]	Jiang & Ye, 2023. IRRA: Cross-Modal Implicit Relation Reasoning and Aligning for Text-to-Image Person Retrieval. CVPR 2023

[3]	Yang et al., 2024. A Pedestrian is Worth One Prompt: Towards Language Guidance Person Re-Identification. CVPR 2024.

**Questions:**

Please refer to Weakness

---

### Official Review · Reviewer_bBQg · 2025-10-28

**Soundness:** 3
**Presentation:** 3
**Contribution:** 2
**Rating:** 2
**Confidence:** 5

**Summary:**

The paper proposes FBA (Foreground and Background Adversarial Person Re-identification), an end-to-end, language-enhanced dual-branch framework that mimics human perception by jointly modeling foreground and background semantics. This addresses the limitation of existing multimodal ReID methods that solely focus on the target foreground, which often leads to feature entanglement between the person and background distractions. FBA introduces an intra-semantic alignment strategy for fine-grained multimodal feature capturing and an inter-semantic adversarial learning strategy with a diversity loss to explicitly distinguish and penalize the feature distance between target and distractor regions, achieving competitive results on both holistic and occluded ReID benchmarks.

**Strengths:**

1. The FBA framework is straightforward and intuitive.
2. The authors conducted extensive experiments on representative datasets spanning four ReID domains.

**Weaknesses:**

1. Attribute-based ReID methods, such as CLIP-ReID and MP-ReID, have demonstrated strong performance in general ReID tasks. MP-ReID [1], similar to FBA, employs large language models (LLMs) to generate additional prompts, achieving 95.50 mAP and 97.70 R@1 on Market1501, respectively, as well as 88.90 mAP and 95.70 R@1 on and DukeMTMC-reID. Although FBA also leverages LLMs to produce supplementary prompts, its performance improvements remain limited.

2. Regarding experimental datasets, Market101 and MSMT are unavailable, and DukeMTMC-reID is discouraged for use due to privacy concerns.

3.  The paper fails to compare its results against recently recognized strong baselines in the occluded ReID domain, specifically CNN-based BPBreID [2] and ViT-based KPR (ECCV 2024) [3].


Reference:

[1] Zhai, Yajing, et al. "Multi-prompts learning with cross-modal alignment for attribute-based person re-identification." Proceedings of the AAAI Conference on Artificial Intelligence. Vol. 38. No. 7. 2024.

[2] Somers, Vladimir, Christophe De Vleeschouwer, and Alexandre Alahi. "Body part-based representation learning for occluded person re-identification." Proceedings of the IEEE/CVF winter conference on applications of computer vision. 2023.

[3] Somers, Vladimir, Alexandre Alahi, and Christophe De Vleeschouwer. "Keypoint promptable re-identification." European Conference on Computer Vision. Cham: Springer Nature Switzerland, 2024.

**Questions:**

Using only foreground and background information to generate two corresponding prompts lacks sufficient granularity for the ReID domain. It is recommended that the authors incorporate part-level prompt learning to further enhance model performance.

**Details Of Ethics Concerns:**

DukeMTMC-reID is discouraged for use due to privacy concerns.

---

### Official Review · Reviewer_LYzf · 2025-10-28

**Soundness:** 2
**Presentation:** 3
**Contribution:** 2
**Rating:** 2
**Confidence:** 5

**Summary:**

This paper proposes an end-to-end framework that jointly models foreground and background information within a dual-branch bidirectional cross-attention feature extraction pipeline. To help the network distinguish between the two domains, we propose an intra-semantic alignment and inter-semantic adversarial learning strategy. The proposed model achieves competitive results on the holistic and occluded ReID datasets.

**Strengths:**

The paper proposes an end-to-end framework that treats both foreground and background semantics as equally important for ReID. The idea is interesting.

**Weaknesses:**

Please refer to Questions.

**Questions:**

•	The paper does not discuss the potential impact of using different large language models (LLMs) for generating textual descriptions of the foreground and background regions. It would be valuable to clarify whether the quality of the generated descriptions influences the performance, and how sensitive the method is to variations in text generation quality.

•	The mechanism for partitioning encoder outputs into foreground and background feature groups is not clearly explained. If the encoder features are shared between the two branches, it may raise concerns about the rationality.

•	The term "FCN" in Figure 3 is ambiguous and lacks a clear definition in the text. Its role and structure should be explicitly described to avoid confusion.

•	The paper excludes the Market-1501 dataset from evaluation, citing its noisy annotations and detection errors that may affect adversarial training stability. However, Market-1501 remains a standard benchmark in ReID research and has been widely used in prior work. To strengthen the validity of the claim, the authors are encouraged to include results on this dataset and provide a more in-depth analysis of how such noise impacts their method compared to existing approaches.

•	In addition to commonly used datasets, the results should also be reported on MSMT17, another widely adopted benchmark in ReID—especially since recent state-of-the-art methods such as CLIP-ReID and PromptSG have evaluated their performance on it. Omitting MSMT17 limits the comprehensiveness of the evaluation.

•	The proposed method employs several implementation tricks, including the sliding-window setting, larger input image size, and smaller stride setting. While these enhancements may contribute to performance gains, they also introduce unfair advantages in comparison with other methods. Notably, on DukeMTMC, CLIP-ReID achieves 83.1% mAP under the sliding-window setting, and TransReID reaches 82.6% mAP with image size 384×128, both outperforming the proposed method (81.7% mAP). Moreover, ablation studies on these design choices are only conducted on CUHK03-NP, leaving the performance on other datasets (e.g., DukeMTMC) under fair configurations unclear.

•	Does a stronger identity-related attention across foreground and background correspond to a larger value of s_j? It is unclear why the weight w_j is defined as 1 − s_j rather than s_j. A justification for this design choice would be helpful.

•	The proposed framework involves the additional text encoder and interaction modules during inference, which may increase model complexity and computational cost. The paper should provide a comparison of training and inference complexity with other methods to assess the practicality of the approach.

**Details Of Ethics Concerns:**

Due to privacy concerns, the DukeMTMC dataset has been withdrawn by the publisher.

---

### Note · Authors · 2025-11-14

I have read and agree with the venue's withdrawal policy on behalf of myself and my co-authors.